# Knowledge, and attitude of service user of intermittent preventive treatment of malaria in pregnancy using sulfadoxine pyrimethamine in the Volta Region of Ghana

**Livingstone Asem** [1,2] *, **Abdul-Gafaru Abdulia** [1], **Patrick Opoku Assuming** [1], **Gordon Abeka-Nkrumah** [1]

1 Department of Public Administration and Health Services Management, Business School, University of Ghana, Accra, Ghana, 2 Department of Health Policy Planning and Management, School of Public Health, University of Health Allied Sciences, Ho, Ghana

* lasem@uhas.edu.gh

## Abstract

### Background

Malaria in pregnancy (MiP) is a condition that can be prevented by using intermittent preventive treatment using Sulfadoxine-pyrimethamine. However, despite all the effort to reduce the consequences of MiP for the woman, the unborn child, and the neonate, the knowledge of Intermittent Preventive Treatment of Malaria in pregnancy using sulfadoxine-pyrimethamine (IPTp-SP) is low in most malaria-endemic countries, including Ghana. Thus, the need to examine knowledge, and attitude of service users of intermittent preventive treatment of malaria in pregnancy using sulfadoxine-pyrimethamine.

### Methods

The study was a cross-sectional survey of two selected districts in the Volta Region of Ghana. The study participants were randomly selected from communities within Nkwanta North and North Tongu District. In all a total of 438 mothers who have delivered in the past 24 months were selected for the study. The women were interviewed using a structured questionnaire and the bivariate and multivariable logistic regression results presented in tables.

### Results

The level of knowledge, and attitude were reported as 45.9% and 58.9% respectively. Knowledge of the service user is determined by the level of education of the women. The attitude of the service user is determined by making 4–7 visits during ANC, Gestational age at booking for ANC is 4–7 weeks, income level between 100 to 999, partner educational level above Middle/JHS/JSS, and age of a partner is above 40 years.

**Data Availability Statement:** All relevant data are within the paper.

**Funding:** The author(s) received no specific funding for this work.

**Competing interests:** The authors have declared that no competing interests exist.

**Abbreviations:** ANC, Antenatal Care; IPTp-SP, Intermittent preventive treatment of malaria in pregnancy using sulfadoxine-pyrimethamine; LMIC, Low- and middle-Income Countries; MIP, Malaria in pregnancy; NHIS, National Health Insurance Scheme.

## Conclusion

The findings from the present studies highlighted important factor such as number of antenatal visits that affect both knowledge of services and attitude to use IPTp-SP. Therefore, a community-based health promotion programmes to help to increase knowledges and improved attitude on timely and regular antenatal attendance to promote the benefit of IPTp-SP should be encouraged.

## Background

Infection with malaria during pregnancy is of significant public health interest in malaria-prevalent regions with steady transmissions, such as tropical Africa. Infections from malaria in the course of pregnancy can cause many unfortunate health effects on pregnant women and their newborns, such as maternal anaemia, underweight babies, and preterm delivery [1–3]. Low birth weight is the highest risk predictor for newborn mortality and a key contributor to the mortality of children under one year [4–6].

Globally, immense progress has been accomplished in the struggle against malaria. From the 2019 global malaria report, it was revealed that in 2018, 228 million malaria cases were reported globally as compared to 251 million cases in 2010. The African region reported about 213 million malaria cases signifying 93% of total cases worldwide. The incidence of malaria infections declined globally from 71 cases per 1000 population at risk in 2010 to 57 cases per 1000 population at risk in 2018, signifying an 18% reduction over the period. In the WHO African region, the malaria incidence rate was relatively high at 216 cases per 1000 population at risk for the year 2017 [7]. Again, in 2017, there were a projected 435,000 deaths from malaria worldwide, with the African zone reporting 93% of all these deaths [8]. Regarding funding, it is estimated that US$ 3.1 billion was expended on malaria control and prevention activities worldwide by leaders of malaria-endemic countries and international stakeholders in 2017, with nearly three-quarters of this amount (about US$2.2 billion) expended in the African zone. Even though financial support for malaria activities has remained moderately fixed since 2010, the amount of financing is far from what is needed to reach the global target ($4.4 billion) of a 40% decrease in malaria morbidity and mortality by the year 2020 [8].

Malaria infection during pregnancy can be prevented using policies such as intermittent preventive therapy (IPTp), proper case management, and the sharing of insecticide-treated bed nets in endemic regions all over the world. In spite of that, just a segment of these pregnant women receives these useful malaria medicines [9].

WHO in 2010 developed guidelines that endorsed microscopy diagnosis or rapid diagnostic test (RDT) for all individuals infected by suspected malaria and this includes pregnant women. Since pregnant women experience other health conditions similar to MiP, proper management of MiP through diagnostics confirmation is very crucial. WHO in a recent policy upgrade, further recommended effective case management, prevention by using long lasting-insecticide bed-nets (LLiNs), and intermittent preventive treatment (IPTp) with sulfadoxine-pyrimethamine for the prevention of malaria disease during pregnancy, delivered through antenatal care (ANC) platform [10–12].

Malaria in pregnancy is projected to be the cause of up to 200,000 infant deaths in Africa yearly. It also causes both preterm births and intrauterine growth retardation. In some cases surviving infants frequently undergo lasting consequences from infection in the womb that hinder their growth and advancement [13, 14]. It was estimated in 2010 that 11.4 million

pregnancies in Sub-Saharan Africa (41% of the estimated 27.6 million live births) could have contracted P. falciparum placental disease at some phases of pregnancy with the absence of malaria prevention intervention during pregnancy [15, 16]. Also, from the global malaria report 2019, it was stated that among 36 African countries that reported on IPTp-SP in 2018, only 31% of qualified pregnant women were given the proposed 3 or more doses of IPTp-SP as compared with 22% in 2017 [8].

Inside Ghana malaria illness during pregnancy constitutes a significant public health challenge. For example, malaria among pregnant women in Ghana represents about 16.8% of hospital admissions and 3.4% of total deaths. Thus, malaria infections among pregnant women constitute a tremendous burden on the country's health system [17].

Ghana, implementing the proposal of the World Health Organisation in 2000, approved fresh malaria management guidelines in 2004 which were reviewed in 2007, and again, in 2012. Accordingly, Ghana changed from the usage of monotherapy to combination treatment using artemisinin-based combination therapy (ACT) and portions of this guidelines were the shift from the practice of weekly Chloroquine chemoprophylaxis to Sulphadoxine–Pyrimethamine as an intermittent treatment for malaria infections during pregnancy [18, 19]. Intermittent preventive therapy during pregnancy (IPTp), is grounded in the belief that each pregnant woman in a malaria-endemic zone has malaria organisms in her blood and/or placenta, whether or not she has symptoms of malaria. Sulphadoxine-pyrimethamine is a single dose antimalarial medicine that has been realized to be very beneficial in stopping malaria during pregnancy and decreasing the magnitude of malaria disease in mothers and babies [20]. In 2014, Ghana reviewed its strategy on IPTp-SP to mirror this revised strategy of the WHO in 2012 [17, 21]. Thus, Ghana has since 2014 implemented the updated policy on IPTp-SP.

Although there are several studies within the general malaria prevention literature that have examined the clinical efficacy of IPTp-SP like van Eijk et al. (2019) and Megnekou et al. (2015) in addressing malaria in pregnancy, there is nonetheless limited information on the knowledge and attitude of the service user of IPTp-SP in Ghana [22, 23]. Hence this study seeks to examine the knowledge and attitude of service users of IPTp-SP in the Volta Region of Ghana.

## Methods and materials

### Study design and setting

A quantitative, cross-sectional study was conducted among women with children less than two years in the Volta Region (Volta and Oti) of Ghana. The Volta Region was one of Ghana's ten (10) administrative regions at the beginning of the study which was later split into sixteen (16) administrative regions. It is situated in the eastern part of the country sharing boundaries in the north with the Northern region, the south with the Gulf of Guinea, in the west with the Volta Lake, and in the east with the Republic of Togo. From the routine service, the region had IPTp3 Coverage of 50.6% in 2019, with Nkwanta South having the lowest coverage of 29% and North Tongu District having the highest coverage of 76.1%. However, comparing the average coverage from 2015 to 2019, Nkwanta North had the lowest coverage of 23.1% with North Tongu having the highest coverage of 59.5%, and both districts selected for the study. The Volta Region was selected because it presents a peculiar and widespread case of the problem under investigation. Based on data from the Policy, Planning, Monitoring and Evaluation Directorate (PPMED) of the Ghana Health Service (GHS), the Volta Region was ranked 10th (last) on the "Performance Gauge" (PG) scale in 2016 and 2017. The Holistic Assessment of the Health Sector Programme of Work 2014 and the DHS report of 2014 mentioned the Volta Region as an area that needs special attention. The region is, therefore, seen as a classical area

for determining the knowledge and attitude of service users of intermittent preventive treatment of malaria in Ghana [24].

## Study population

The study population includes all women who gave birth twenty-four months preceding the survey from the communities within the region expected to have used antenatal services. The survey date was 1ˢᵗ June 2020 to 10ᵗʰ July 2020. This excludes all women who gave birth long before the past 24 months in the Volta Region.

## Variables

This study included the following variables that have been linked in theory and empirically to malaria in pregnancy and the use of IPTp-SP. The outcome/dependent variables were knowledge of 3+ doses of IPTp-SP by women exposed and the attitude of exposed women toward IPTp-SP intervention after policy upgrade. The independent/explanatory variables include the demographic characteristics of the respondents such as age, marital status, educational level of woman, occupation of the woman, religion, ethnic group, partner education, partner occupation, partner age, duration of marriage, monthly household income, household assert, means of transportation to the nearest health facility, estimated travelled time to the nearest health facility, estimated distance to the nearest health facility, health insurance enrollment, and health insurance validity status, gestational age booked for antenatal, gestational age for IPTp-SP, gravida, parity, number of live births, and number of antenatal visits. The questionnaire used in the study is provided in the supporting information (S1 Checklist).

## Sample size determination and sampling

The formula for sample size determination based on the Yamane (1967) approach is stated as follows:

$$n = \frac{N}{1 + N(e)^2} \tag{1}$$

Where n is the sample size, N is the population size, e is the level of precision. Yamane's (1967) approach to sample size determination assumes that the population of the study is known and is finite. The sample frame (target) population is estimated to be 76,341. The appropriate sample size is determined based on Yamane's (1967) approach at the precision of 5% as presented below:

$$n = \frac{76,341}{1 + 76,341(0.05)^2} = 398 \tag{2}$$

The required sample size that was selected is therefore 398. Given the possibility that not all the target respondents were reached (non-response), 10% of the obtained sample size, four hundred and thirty-eight (438), were added to compensate for persons that the researcher was unable to reach.

The sampling procedure was adapted from the 2014 Ghana Demographic and Health Survey (GDHS) and followed a two-stage sample design intended to allow estimates of key indicators for urban, and rural areas in the two selected districts. The first stage using probability proportional to the size and then thirty households per cluster scenario [25], the researcher selected eight (8) clusters for North Tongu and six (6) clusters for Nkwanta North, giving a total of fourteen(14) clusters for the study area with five (5) urban and nine (9) rural. The

second stage of sampling involved the selection of the sampling unit for enumeration after listing all members of each household in the enumeration area. All the eligible women were sampled based on the population proportion to the size of the eligible population contribution after the listing of all the members of the households in the clusters and then a random selection of the eligible population was done using Stata to select the respondents. In all, 438 women were selected at random for the interview. However, those that were not available at the time of the enumeration were replaced by the next in the randomized order.

## Data collection and quality management

The study used a structured questionnaire for data collection. The questionnaire solicited data on the demographics and use of IPTp-SP. Data on the number of antenatal visits, number of doses of IPTp-SP, parity, and gravida was obtained from the questionnaire and triangulated by their maternal health records. Data was captured using Kobo collect app, data extracted as a Microsoft Excel file, and exported to Stata 14.1 for analysis. Data cleaning was done at every stage to ensure that we had good-quality data for analysis.

## Data analysis

In the analysis, the continuous independent variables were categorized into intervals range as required whilst the categorical variables such as sex, educational level, ethnicity, etc. retained their categorization during analysis. To ascertain the wealth index, the researcher had to first conduct a principal component analysis of the household assets and possessions to establish which variable should be included in the model to determine the wealth of the household. In the study, data was collected on household assets and possessions which may be correlated in an unknown and complex way. Thus, when a set of variables are correlated in a complex and unknown way along several dimensions, Principal Component Analysis (PCA) is employed to reduce these variables by assessing which variables behave in a similar manner. Based on the variables and their relationship to each other, PCA creates a new set of variables called the principal component. The household assets and possessions collected during the study were assigned weights based on the PCA and the resulting scores were standardized to a standard normal distribution. Based on these aggregate scores, individuals and their household wealth were put into five quintiles. To determine the level of knowledge, the researcher had to first conduct a principal component analysis (PCA) of the knowledge questions to determine which variables should be included in the model to determine the level of knowledge. In the study, data were collected on 11 items that may be correlated in an unknown and complex way. Thus, when a set of variables are correlated in a complex and unknown way along several dimensions, principal component analysis (PCA) is employed to reduce these variables by assessing which variables behave in a similar manner. Based on the variables and their relationship to each other, PCA creates a new set of variables called the principal component [26]. The answers to the knowledge questions collected during the study were assigned weights based on the PCA and the resulting scores were standardized to standard normal distribution. Information on knowledgeability that were binary variables (which elicited "No" and "Yes" responses) were recorded as either 0 or 1 where 0 means the service user does not know whilst 1 represents the service user knowing. After the PCA, variables with eigenvalue (a measure of its power to explain variation between participants) above one, were put together to determine the knowledgeability level of the individuals. Usually, a factor is considered important and worthy of inclusion in the scale if its eigenvalue exceeds the threshold of >1 [27]. All 4 items were retained in the final model to determine knowledge. Knowledge was measured by calculating the mean score of 4 items for the service

user and categorized as knowledgeable (if participants scored $\geq$ a mean score of the correctly answered questions) or not knowledgeable (if participants scored <mean score of the correctly answered questions). Based on these weighted (aggregated) scores of individuals were placed into being knowledgeable/not knowledgeable. This model has previously been used by Kassahun et al in similar study to classify knowledge of study participants and was found to be a statistically significant way of classifying knowledge of study participants [28]. A Cronbach Alpha index was computed to test the reliability of this measurement approach [29]. This gave rise to Cronbach Alpha of 0.6 which is within the acceptable level of reliability [30].

To determine the attitude, the researcher had to first conduct a principal component analysis (PCA) of the attitude questions to determine which variables should be included in the model to determine the attitude. In the study, data was collected on 16 items that may be correlated in an unknown and complex way. Thus, when a set of variables are correlated in a complex and unknown way along several dimensions, principal component analysis (PCA) is employed to reduce these variables by assessing which variables behave in a similar manner. Based on the variables and their relationship to each other, PCA creates a new set of variables called the principal component [26]. The answers to the attitude questions collected during the study were assigned weights based on the PCA and the resulting scores were standardized to standard normal distribution. Usually, a factor is considered important and worthy of inclusion in the scale if its eigenvalue exceeds the threshold of >1 [27]. All 6 items were retained in the final model to determine attitude. The 6 items on attitudinal questions were computed to obtain total scores; then, the mean score was calculated to categorize as having a good attitude (if participants scored $\geq$ mean score) or poor attitude (if participants scored < mean score). However, there was a regrouping of the variables into *yes* and *no* for easy computation such that *agree* and *agree strongly* as "yes" and *disagree* and *disagree strongly* as no with *yes* for good attitude and "*no*" for poor attitude [28]. A Cronbach Alpha index was computed to test the reliability of this measurement approach [29]. This gave rise to Cronbach Alpha of 0.5 which is within the acceptable level of reliability [30]. Chi-Square test of associations was used to check for dependence between the sociodemographic variables and the outcome variables. All independent variables with p<0.05 in the bivariate analysis were included in the multivariable logistic regression to further examine the association between the outcome and each independent variable while controlling the effect of other explanatory variable by calculating adjusted odds ratios (AOR). The level of significance used was 5% (0.05), two tailed at 95% confidence interval (CI). The goodness of fit of the model was tested by using the criterion-based method, usually known as the "Leaps-and-Bounds algorithm".

To investigate the effect of the predictor variable on the outcome variables, binary logistic regression analysis was done and a p-value < 0.05 was considered statistically significant.

## Ethical approval and consent to participate

Ethical clearance was obtained from Ghana Health Service's Ethical Review Committee (ERC) with protocol ID No: **GHS-ERC 005/12/19** through the University of Ghana, Business School. The researcher obtained written informed consent and assents were obtained from each study participants whose age was 18 years and above and below 18 years respectively. Willingness to participate in the study and parental permission was confirmed by signing or finger print on the informed consent form. Ethical approval letter and informed consent form is provided in the OTHER (ERC_clearance) and (Informed Consent).

## Results

### Sociodemographic characteristics of the service users

The average age of the respondents was 29 years (SD:6.1) with 211 (54.8%) of them falling within the 20-29-year group. Fifty-four percent (54.1%) of the respondents, had lower than JHS/JSS/Middle education. Christians, 353 (80.6%), formed the majority of the religious groupings. Seventy-eight percent (77.6%), were married. 155 of the respondents, representing thirty-five percent (35.39%), were Farmers, with 41 (9.36%) of them being unemployed. The most dominant ethnic group was Ewes, who constituted 54.79% of the respondent. Most of the partners of the women, 196 (44.7%), were between the ages of 30 and 39 with the mean age of 35 years (SD:7.6). One hundred and eighty-nine of the partners of the respondents had less than JHS/JSS/Middle educational level and 187 of them, representing forty-three percent (42.69%), were farmers. Most of the respondents, 242 (55.25%), had been married for 3 to 9 years with an average duration of 7 years (SD:4.9). An average household income for the respondents was GH ¢349 per month with a majority of them, 243 (55.48%), earning between 100 to 999 Ghana cedis per month. The highest wealth index quintile was the second with 90 (20.55%) of the women in this category. This was computed for the list of assets like owning a mobile phone, a television set, radio, bed, farmland, means of transport, house, etc. The majority of the respondents, 245 (55.9%), resided in the rural areas of the region with only 193 (44.1%) in the urban areas. 210 of the respondents, representing forty-eight percent (47.95%), walked to the nearest health facility during antenatal care with the average estimated travel time and distance being 32 minutes (SD:21.0) and 8km (SD:9.1) respectively. A high number of the respondents, 282 (64.68%), booked for antenatal care in the first trimester. An overwhelming majority of them, 408 (93.15%), had enrolled on the National Health Insurance Scheme. However, only 221 (50.46%) of them were active to access free services. The majority of the women, 237 (54.1%), had a parity of 0 to 2 pregnancies, the average gravida was 3 (SD:1.9) pregnancies and the average number of live births was 3 (SD:1.8) children. The mean number of antenatal care (ANC) attendance was 5 (SD:2.6) visits per client, with 262 (59.82%) of them getting the 3+ doses of IPTp-SP while 44 (10.1%) of the respondent did not receive any dose of IPTp-SP. 221 (50.46%) were booked for IPTp-SP before the 17[th] week as shown in Table 1 below.

### Knowledge on IPTp-SP

Three Hundred and eighty-nine (389) of the respondents, representing 88.8%, knew that malaria is caused by mosquito bites. A good number of the service users (303 representing 69.2%) knew that the main signs and symptoms of malaria are high body temperature. Less than half of the women 217 (49.54%) did not know the effect of malaria on pregnant women. More than half of the respondents (241, representing 55%) did not know the effect of malaria on the unborn child. The majority of the women 402 (91.8%) knew the various methods of preventing malaria. Most of the women, 318 (73.44%), knew that the purpose of taking IPTp-SP is to prevent malaria; however, 399 (91.1%) did not know the number times to receive IPTp-SP. Also, most of the respondents, 402 (91.78%), got to know about IPTp-SP for the first time from their health providers. The average knowledge score of 6.14 (SD: 2.1) with 237 (54.1%) having poor knowledge or not knowledgeable on malaria in pregnancy as shown in Table 2 below.

### Sociodemographic variables and knowledgeability on IPTp-SP

From the cross-tabulations in Table 3 below, only 3 variables out of 25 were identified as having an association with the knowledge of the respondents (p<0.05). These variables are

**Table 1. Background characteristics of IPTp-SP service users.**

| Item | Number N = 438 | Percentage (%) |
|---|---|---|
| **Age group(women)** | | |
| Below 20 | 20 | 4.6 |
| 20–29 | 211 | 48.2 |
| 30+ | 207 | 47.2 |
| **Educational level (woman)** | | |
| <Middle/JHS/JSS | 237 | 54.1 |
| Middle/JSS/JHS | 134 | 30.6 |
| >Middle/JHS/JSS | 67 | 15.3 |
| **Religious affiliation (woman)** | | |
| Christians | 353 | 80.6 |
| Muslim | 23 | 5.3 |
| Others | 62 | 14.2 |
| **Marital Status (woman)** | | |
| Co-habitation | 60 | 13.7 |
| Married | 340 | 77.6 |
| Others | 38 | 8.7 |
| **Occupation (woman)** | | |
| Farmer/Agriculture | 155 | 35.4 |
| Government worker | 19 | 19 |
| Trader/Artisans | 223 | 36.2 |
| Unemployed | 41 | 9.4 |
| **Ethnic Group (woman)** | | |
| Ewe | 240 | 54.79 |
| Guan | 9 | 2.05 |
| Konkomba | 137 | 31.28 |
| Other | 52 | 11.87 |
| **District** | | |
| Nkwanta north | 182 | 41.55 |
| North Tongu | 256 | 58.45 |
| **Partner Age** | | |
| 20–29 | 104 | 43.2 |
| 30–39 | 196 | 44.7 |
| 40+ | 138 | 31.5 |
| **Educational level (Partner)** | | |
| <Middle/JHS/JSS | 189 | 43.2 |
| Middle/JSS/JHS | 123 | 28.1 |
| >Middle/JHS/JSS | 126 | 28.8 |
| **Occupation (Partner)** | | |
| Farmer/Agriculture | 187 | 42.69 |
| Government workers | 56 | 12.79 |
| Trader/Artisans | 190 | 6.16 |
| Others | 5 | 19.86 |
| **Duration of marriage** | | |
| <3 | 82 | 18.72 |
| 3–9 | 242 | 55.25 |
| 10+ | 114 | 26.03 |

*(Continued)*

**Table 1.** (Continued)

| Item | Number<br>N = 438 | Percentage<br>(%) |
|---|---|---|
| **Monthly Household Income** | | |
| <100 | 156 | 35.62 |
| 100–999 | 243 | 55.48 |
| 1000+ | 39 | 39 |
| **Wealth Index** | | |
| Lowest | 88 | 20.09 |
| Second | 90 | 20.55 |
| Middle | 85 | 19.41 |
| Fourth | 88 | 20.09 |
| Highest | 87 | 19.86 |
| **Residence** | | |
| Urban | 193 | 44.06 |
| Rural | 245 | 55.94 |
| **Means of Transport** | | |
| Boat | 1 | 0.23 |
| Motor bike | 164 | 37.44 |
| Vehicle | 63 | 14.38 |
| Walking | 210 | 47.95 |
| **Estimated travel time to the nearest health facility(One-Way)** | | |
| <30mins | 276 | 60.96 |
| 31-60mins | 156 | 35.62 |
| 1hr+ | 15 | 3.42 |
| **Estimated distance to the nearest health facility** | | |
| <3km | 143 | 32.72 |
| 3-5km | 105 | 24.03 |
| 6+km | 189 | 43.25 |
| **Gestational Age at Booking for ANC** | | |
| 1st Trimester | 282 | 64.68 |
| 2nd Trimester | 131 | 30.05 |
| 3rd Trimester | 23 | 5.28 |
| **NHIS enrollment Status** | | |
| Not enrolled | 30 | 6.85 |
| Enrolled | 408 | 93.15 |
| **Validity of NHIS card** | | |
| Not active/never enrolled | 217 | 49.54 |
| Active | 221 | 50.46 |
| **Number of pregnancies(gravida)** | | |
| <3 | 208 | 47.49 |
| 3–4 | 137 | 31.28 |
| 5+ | 93 | 21.23 |
| **Parity** | | |
| 0–2 | 237 | 54.1 |
| 3–4 | 124 | 28.3 |
| 5+ | 77 | 17.6 |
| **Livebirths** | | |
| 0–2 | 236 | 53.9 |

(*Continued*)

**Table 1.** (Continued)

| Item | Number N = 438 | Percentage (%) |
|---|---|---|
| 3–4 | 126 | 28.77 |
| 5+ | 76 | 17.35 |
| **Number of ANC Visits** | | |
| <4 | 100 | 22.83 |
| 4–7 | 264 | 60.27 |
| 8+ | 74 | 16.89 |
| **Doses of SP Received** | | |
| No dose | 44 | 10.05 |
| 1–2 doses | 123 | 30.14 |
| 3+ doses | 262 | 59.82 |
| **Gestational age at booking for IPTp-SP** | | |
| <17 wks | 221 | 50.46 |
| 17-24wks | 145 | 33.11 |
| 25wks | 72 | 16.44 |

estimated distance to the nearest health facility, gestational age at booking for antenatal and antenatal visits. Other variables like the household income, parity, gravida, etc. had no association with the knowledge of the service user on IPTp-SP use as shown in Table 3 below:

## Crude and adjusted associations between sociodemographic variables and knowledge of IPTp-SP

From the bivariate analysis in Table 3 above, 3 variables out of 25 variables had relationships with the knowledge ($p<0.05$). Using the criterion-based method, usually known as "Leaps-and-Bounds algorithm", the optimal model selected among the nested models was model 4 (AIC = 572.7595). The predictors selected are: number of antenatal care visits, means of transport, livebirth and parity. The multivariable logistic regression analysis from Table 4 below shows that the number of antenatal visits was the only variable associated with knowledge IPTp-SP. Overall, those who attended antenatal clinic 4–7 visits have 2.4 (CI:1.4–4.4) time higher knowledge than those who attend less than 4 visits. Similarly, among women who had 8 and more visits have 7.6(CI: 2.6–22.1) times higher odds of being knowledgeable in malaria in pregnancy thus improve uptake as compared with those who had less than 4 visits.

## Service users' attitude towards the uptake of IPTp-SP

From Table 5 below, 428 (97.7%) of the service users believed that early booking for antenatal care is good; however, 409 (93.4%) of the respondents said they booked for ANC before the 12th week. Almost all, 414 (94.5%), of the service users believed that intermittent preventive treatment of malaria in pregnancy using Sulfadoxine. The majority of the service users, 297 (67.8%), were of the view that good staff attitude encouraged the continuous utilization of the IPTp-SP. On the issue of waiting time at the health facility, a minority of the clients, 198 (45.2%), agree it discourages then from going to the health facility. The greater number of service users, 425 (97%), said there was privacy at the consulting room. Most of the respondents agreed they would seek regular ANC attendance during pregnancy. On the issue of planned pregnancy, 204 (47%) did not plan before getting pregnant. The majority, of the service users, 330 (75.3), agreed that they could meet transportation costs to attend ANC. 225 of the women,

**Table 2. Service users' knowledge on IPTp-SP.**

| Item | Number N = 438 | Percentage (%) |
|---|---|---|
| **Cause of malaria** | | |
| Not knowledgeable | 49 | 11.19 |
| Knowledgeable | 389 | 88.8 |
| **Signs and symptoms of malaria** | | |
| Not knowledgeable | 135 | 30.8 |
| Knowledgeable | 303 | 69.2 |
| **Effect of malaria on pregnant women** | | |
| Not knowledgeable | 217 | 49.5 |
| Knowledgeable | 221 | 50.5 |
| **Effect of malaria on unborn child** | | |
| Not knowledgeable | 241 | 55 |
| Knowledgeable | 197 | 45 |
| **Malaria Prevention method** | | |
| Not knowledgeable | 36 | 8.2 |
| Knowledgeable | 402 | 91.8 |
| **Purpose of IPTp-SP** | | |
| Not knowledgeable | 115 | 26.6 |
| Knowledgeable | 318 | 73.4 |
| **Time to start IPTp-SP** | | |
| Not knowledgeable | 181 | 42.2 |
| Knowledgeable | 248 | 57.81 |
| **Know about number of tablets** | | |
| Not knowledgeable | 173 | 39.5 |
| Knowledgeable | 265 | 60.5 |
| **Number of times to Receive SP** | | |
| Not knowledgeable | 399 | 91.1 |
| Knowledgeable | 39 | 8.9 |
| **Interval between doses of SP** | | |
| Not knowledgeable | 130 | 28.04 |
| Knowledgeable | 308 | 71.96 |
| **Source of Information on SP** | | |
| Health provider | 402 | 91.78 |
| Radio | 5 | 1.14 |
| TV | 4 | 0.91 |
| Pregnant women | 13 | 2.97 |
| Others | 14 | 3.20 |
| **Knowledge Score** | | |
| Not Knowledgeable | 237 | 54.1 |
| Knowledgeable | 201 | 45.9 |

representing 51.4%, mentioned that they had no power to make their own decisions concerning the use of antenatal services, although most of them, 289 (66%), got the support of their partners to attend antenatal. Most of the service users, 425 (97%), described the service provided by the service users as good. The mean attitude score was 11.7 (SD:2.07) with 258 (58.9%) of them with good attitude whilst 180 (41.1%) with poor attitude to healthcare.

**Table 3. Sociodemographic variables and knowledgeability on IPTp-SP.**

| Variables | Number N = 438 | Not knowledgeable | Knowledgeable | Ch12 | p-value |
|---|---|---|---|---|---|
| **Age group(women)** | | | | 0.35 | 0.855 |
| Below 20 | 20 | 2.9 | 4 | | |
| 20–24 | 211 | 50.3 | 49.9 | | |
| 25–29 | 207 | 46.8 | 46.1 | | |
| **Educational level (woman)** | | | | 38.8 | 0.068 |
| <Middle/JHS/JSS | 166 | 61.2 | 51.3 | | |
| Middle/JSS/JHS | 71 | 32.9 | 21.6 | | |
| >Middle/JSS/JHS | 134 | 5.9 | 27.1 | | |
| **Religion** | | | | 1.04 | 0.754 |
| Christians | 353 | 88.3 | 84.9 | | |
| Muslims | 23 | 2.7 | 34.5 | | |
| Others | 62 | 9 | 11.6 | | |
| **Marital Status** | | | | 2.8 | 0.691 |
| Co-habitation | 60 | 10.5 | 14.7 | | |
| Married | 340 | 84.4 | 78 | | |
| Others | 38 | 5.1 | 7.2 | | |
| **Occupation (woman)** | | | | 30.2 | 0.158 |
| Farmer/Agriculture | 155 | 56.7 | 43.9 | | |
| Government Worker | 19 | 0.4 | 11.3 | | |
| Trader/Artisans | 223 | 36.6 | 34.5 | | |
| Unemployed | 41 | 6.3 | 10.3 | | |
| **Ethnic Group (woman)** | | | | 21.6 | 0.251 |
| Ewe | 240 | 43.8 | 36.7 | | |
| Guan | 9 | 0.3 | 8.9 | | |
| Konkomba | 137 | 45.1 | 42.4 | | |
| Other | 52 | 10.9 | 12 | | |
| **District** | | | | 1.6 | 0.705 |
| Nkwanta North | 182 | 53.9 | 60.1 | | |
| North Tongu | 256 | 46 | 39.9 | | |
| **Partner Age** | | | | 2.3 | 0.603 |
| 20–29 | 104 | 24.2 | 19.9 | | |
| 30–39 | 196 | 42.9 | 49.9 | | |
| 40+ | 138 | 32.9 | 30.2 | | |
| **Educational level (Partner)** | | | | 31.7 | 0.118 |
| <Middle/JSS/JHS | 189 | 57.6 | 41.9 | | |
| Middle/JSS/JHS | 123 | 28.27 | 20.7 | | |
| >Middle/JSS/JHS | 126 | 14.1 | 37.4 | | |
| **Occupation (Partner)** | | | | 8.27 | 0.379 |
| Farmer/Agriculture | 187 | 55.2 | 43.5 | | |
| Government Worker | 56 | 10.1 | 17.4 | | |
| Trader/Artisans | 190 | 34.3 | 39 | | |
| Other | 5 | 0.5 | 0.1 | | |
| **Duration of marriage** | | | | 17.1 | 0.077 |
| <3 | 82 | 9.3 | 22.4 | | |
| 3–9 | 242 | 62.2 | 59.3 | | |
| 10+ | 114 | 28.6 | 18.3 | | |
| **Monthly Household Income** | | | | 17.6 | 0.187 |

*(Continued)*

**Table 3.** (Continued)

| Variables | Number N = 438 | Not knowledgeable | Knowledgeable | Ch12 | p-value |
|---|---|---|---|---|---|
| <100 | 156 | 51.7 | 32.8 | | |
| 100–999 | 243 | 43.4 | 63.6 | | |
| 1000+ | 39 | 4.9 | 3.6 | | |
| **Wealth Index** | | | | 23.4 | 0.251 |
| Lowest | 88 | 20.4 | 12.7 | | |
| Second | 90 | 19.6 | 18.4 | | |
| Middle | 85 | 26.6 | 16.3 | | |
| Fourth | 88 | 25.5 | 32.1 | | |
| Highest | 87 | 7.9 | 20.5 | | |
| **Residence** | | | | 4.2 | 0.43 |
| Urban | 193 | 83.5 | 75.6 | | |
| Rural | 245 | 16.5 | 24.4 | | |
| **Means of Transport** | | | | 1.6 | 0.731 |
| Boat | 1 | 0.1 | 0 | | |
| Motor bike | 164 | 31.9 | 32.4 | | |
| Vehicle | 63 | 9.6 | 6.3 | | |
| Walking | 210 | 58.6 | 61.2 | | |
| **Estimate travel time to nearest health facility** | | | | 3.4 | 0.183 |
| <30mins | 267 | 64.3 | 72.6 | | |
| >29 | 171 | 35.7 | 27.4 | | |
| **Estimated distance to nearest health facility** | | | | 41.2 | 0.017 |
| <5km | 184 | 26.9 | 43.1 | | |
| 5-9km | 162 | 62.7 | 32.4 | | |
| 10+km | 92 | 10.4 | 24.5 | | |
| **Gestational Age at Booking for ANC** | | | | 31 | 0.002 |
| 1st Trimester | 282 | 54.2 | 79.8 | | |
| 2nd Trimester | 131 | 41 | 18.1 | | |
| 3rd Trimester | 23 | 4.8 | 2 | | |
| **NHIS enrollment status** | | | | | |
| Not enrolled | 30 | 5.9 | 10.3 | | |
| Enrolled | 408 | 94.1 | 89.7 | | |
| **Validity of NHIS card** | | | | 1.37 | 0.631 |
| Not active/not enrolled | 217 | 49 | 43.4 | | |
| Active | 221 | 51 | 56.6 | | |
| Gravidity | | | | 10.1 | 0.488 |
| <3 | 208 | 38.3 | 52.5 | | |
| 3–4 | 137 | 32.4 | 28.6 | | |
| 5+ | 93 | 29.3 | 18.9 | | |
| **Parity** | | | | 4.4 | 0.589 |
| 0–2 | 237 | 52.5 | 55.6 | | |
| 3–4 | 124 | 21.9 | 26.2 | | |
| 5+ | 77 | 21.9 | 18.2 | | |
| **Number of Livebirths** | | | | 4.9 | 0.557 |
| 0–2 | 236 | 51.7 | 56.8 | | |
| 3–4 | 126 | 22.7 | 25.9 | | |
| 5+ | 76 | 25.7 | 17.5 | | |
| **Number of ANC Visits** | | | | 27.6 | 0.004 |

(*Continued*)

**Table 3.** (Continued)

| Variables | Number N = 438 | Not knowledgeable | Knowledgeable | Ch12 | p-value |
|---|---|---|---|---|---|
| <4 | 100 | 31.4 | 13.1 | | |
| 4–7 | 264 | 64.2 | 73.1 | | |
| 8+ | 74 | 4.4 | 13.5 | | |
| **Gestational age at booking for IPTp-SP** | | | | 5.25 | 0.564 |
| <17 wks | 221 | 68.6 | 59.3 | | |
| 17-24wks | 145 | 21.5 | 24.5 | | |
| 25wks | 72 | 9.9 | 16.2 | | |

N: represents frequency, p<0.05

## Crosstabulation of IPTp-SP service users' attitude and sociodemographic

From the table below, 10 out of 25 variables were significant (p<0.05). These variables are ethnic group, means of transport, NHIS card active, household income, number of ANC visits, districts, partners age, partner's occupation, marital status and partner's education. 87.4% of service users with good attitude were married, 55.5% of them were Konkomba by tribe, 73% of them were from the Nkwanta North District, 61.2% of them had education below JSS/JHS level, most of them (61.7%) had partners who were farmers, 57.8% had a monthly household income of less than 100 Ghana cedis, 66.4% of them walked to the health facility, 59.7% of them had valid NHIS card and only 9.3% of them attended ANC 8 times or more. These are shown in the Table 6: below:

## Crude and adjusted associations between variables and attitude of service users

From the bivariate analysis in Table 6 above, 10 out of 25 variables had relationships with the attitude and health-seeking behaviour (p<0.05). These variables are marital status, ethnic group, partners age, education and occupation, household income, means of transport, valid

**Table 4.** Crude and adjusted associations between sociodemographic variables and knowledge.

| Variables | Crude OR | 95%CI | p-value | Adjusted OR | 95%CI | p-value |
|---|---|---|---|---|---|---|
| **Number of ANC Visits** | | | | | | |
| <4 | 1 | | | 1 | | |
| 4–7 | 2.7 | 1.1–6.6 | 0.026 | 2.4 | 1.4–4.4 | 0.004 |
| 8+ | 7.4 | 2.7–19.97 | 0.001 | 7.6 | 2.6–22.1 | 0.001 |
| **Travel time** | | | | | | |
| <31 | 1 | | | 1 | | |
| >30 | 0.8 | 0.4–1.2 | 0.184 | 0.8 | 0.4–1.3 | 0.29 |
| **Travel distance** | | | | | | |
| <5km | 1 | | | 1 | | |
| 5–9 | 0.3 | 0.1–0.8 | 0.015 | 0.3 | 0.1–0.7 | 0.011 |
| 10+ | 1.4 | 0.4–5.5 | 0.543 | 1.9 | 0.5–7 | 0.275 |

**Crude odds ratio (OR)**: odds ratio of one independent variable predicting the dependent variable, **Adjusted odds ratio (OR)**: holds other relevant variables constant and provides the odds ratio from the potential variable of interest which is adjusted for the other independent variables included in the model. CI: confidence interval, p<0.05.

**Table 5. Service users' attitude and health seeking behaviour for IPTp-SP.**

| Item | Number N = 438 | Percentage(%) |
|---|---|---|
| **Do you know that early booking for ANC is good** | | |
| No | 10 | 2.3 |
| Yes | 428 | 97.7 |
| **Will you book for ANC before the 12th week** | | |
| No | 29 | 6.6 |
| Yes | 409 | 93.4 |
| **Do you believe IPTp-SP is good for the unborn baby** | | |
| No | 24 | 5.5 |
| Yes | 414 | 94.5 |
| **Did you know follow-up visit is good for mother and baby** | | |
| No | 12 | 2.7 |
| Yes | 426 | 97.3 |
| **Did staff attitude encourage the continuous utilisation of ANC** | | |
| No | 141 | 32.2 |
| Yes | 297 | 67.8 |
| **Did you wait for a long time at the health facility** | | |
| No | 240 | 54.8 |
| Yes | 198 | 45.2 |
| **Was there privacy and confidentiality in the consulting room** | | |
| No | 13 | 3.0 |
| Yes | 425 | 97.0 |
| **Will you seek regular ANC attendance during pregnancy** | | |
| No | 61 | 13.9 |
| Yes | 377 | 86.1 |
| **Was your last pregnancy planned** | | |
| No | 204 | 47.0 |
| Yes | 234 | 53.4 |
| **Do you wait for your baby to move before attending ANC** | | |
| No | 315 | 71.9 |
| Yes | 123 | 28.1 |
| **Are you able to meet transportation cost to ANC** | | |
| No | 108 | 24.7 |
| Yes | 330 | 75.3 |
| **When feeling well and do not have any problem, will you attend ANC Regularly** | | |
| No | 200 | 45.7 |
| Yes | 238 | 54.3 |
| **Do you have the power to make your own decision on ANC** | | |
| No | 225 | 51.4 |
| Yes | 213 | 48.6 |
| **Do you get your partner's support and approval** | | |
| No | 149 | 34.0 |
| Yes | 289 | 66.0 |
| **Will you take five or more ANC visits during your next pregnancy** | | |
| No | 121 | 27.6 |
| Yes | 317 | 72.4 |
| **Can you describe quality and content of service as good during last pregnancy** | | |

*(Continued)*

**Table 5.** (Continued)

| Item | Number N = 438 | Percentage(%) |
|---|---|---|
| No | 13 | 3.0 |
| Yes | 425 | 97.0 |
| **Attitude Score** | | |
| Poor | 180 | 41.1 |
| Good | 258 | 58.9 |

NHIS card, number of ANC visits and districts. The criterion-based method, usually known as "Leaps-and-Bounds algorithm", was used with AIC = 498.0475. The predictors selected are: number of antenatal care visits, district of residence, religion and means of transport. Multi-variable logistic regression analysis from Table 7 below shows that marital status, religion, leaving in a district and number of antenatal visits were associated with attitude. Overall, the women with good attitudes living in North Tongu had 0.1 (CI:0.02–0.18) times less good attitudes than their colleagues in Nkwanta North. For the number of antenatal visits, the women with good attitudes had 17.2 (CI:3.4–87.3) times higher good attitudes as compared with those who attended antenatal less than 4 times. The married women had 0.8(CI:0.3–2.3) times less good attitude as compared with single mothers as shown in Table 7 below.

## Discussion

### Discussion of service users' knowledge, and attitude

This study aimed to assess the factors associated with knowledge, and attitude among pregnant women in the Volta Region of Ghana after the policy update by WHO in 2012. From the study, the overall knowledge, and attitude of the women were 45.9%, and 58.9% for knowledgeable, and good attitudes respectively. Among the covariates analysed for knowledge, and attitude for the service users of IPTp-SP, only the number of antenatal visits was associated with knowledge and attitude. Among the women knowledgeable in malaria in pregnancy was associated with women who did 8 and more visit to ANC. Thus, the better one's knowledge in malaria in pregnancy, the more likely one will attend antenatal care.

It is worth stating, with regard to attitude, the district of residence was associated with good attitude of the service users. Thus, women from the North Tongu District have 0.1 times less good attitude compared with those from Nkwanta North.

The discrepancies in knowledge and attitude were due to many factors, which are confirmed in an earlier study in Nigeria where poor knowledge of SP as a medicine for IPTp was observed [31]. The low knowledge reported in the study could be the reasons for the low uptake of IPTp-SP because pregnant women do not know the benefits of IPTp-SP during pregnancy. This is supported by the fact that only those who had knowledge about the benefits of IPTp-SP were more likely to receive 3 or more doses of SP. This finding stands in sharp contrast to a study in northeast Tanzania, where pregnant women were generally aware of SP as recommended medicine for IPTp [32]. This poor knowledge on IPTp-SP has implications for malaria prevention in pregnancy since pregnant women will not demand SP during their antenatal visits; this hinders meeting the global target of 80%. A study conducted in Tanzania also reported that the majority of respondents believed that anti-malaria medicines are harmful to a pregnant woman and her unborn child [33].

The findings from the current study show that the majority of the pregnant women visited health facilities. This clearly shows that there is an opportunity to educate these women on

**Table 6. Weighted crosstabulation of service users' attitude and health seeking behaviour by sociodemographic variables.**

| Variables | Number N = 438 | Poor | Good | Ch12 | p-value |
|---|---|---|---|---|---|
| **Age group(women)** | | | | 7.9 | 0.357 |
| Below 20 | 20 | 5.9 | 2.1 | | |
| 20–24 | 211 | 54.8 | 47.62 | | |
| 25–29 | 207 | 39.3 | 50.3 | | |
| **Educational level (woman)** | | | | 17.44 | 0.065 |
| <Middle/JHS/JSS | 166 | 48.8 | 61.2 | | |
| Middle/JSS/JHS | 71 | 40.1 | 21.5 | | |
| >Middle/JSS/JHS | 134 | 11.1 | 17.3 | | |
| **Religious affiliation (woman)** | | | | 0.006 | 0.998 |
| Christians | 353 | 86.9 | 86.9 | | |
| Muslim | 23 | 2.9 | 3.1 | | |
| Others | 62 | 10.1 | 10.2 | | |
| **Marital Status (woman)** | | | | 19.4 | 0.011 |
| Co-habitation | 60 | 18.6 | 9.1 | | |
| Married | 340 | 70.7 | 87.4 | | |
| Others | 38 | 10.8 | 3.5 | | |
| **Occupation (woman)** | | | | 4.2 | 0.656 |
| Farmer/Agriculture | 155 | 47.9 | 52.8 | | |
| Government Workers | 19 | 3.4 | 6.1 | | |
| Sales & Service/Trader | 223 | 41.5 | 32.6 | | |
| Unemployed | 41 | 7.1 | 8.5 | | |
| **Ethnic Group (woman)** | | | | 80.1 | 0.001 |
| Ewe | 240 | 68.6 | 25.8 | | |
| Guan | 9 | 0 | 6.2 | | |
| Konkomba | 137 | 22.2 | 55.5 | | |
| Other | 52 | 9.2 | 125 | | |
| **District** | | | | 89.24 | 0.000 |
| Nkwanta North | 182 | 26 | 73 | | |
| North Tongu | 256 | 74 | 27 | | |
| **Partner Age** | | | | 13.1 | 0.036 |
| 20–29 | 104 | 29.5 | 18.4 | | |
| 30–39 | 196 | 48.7 | 44.6 | | |
| 40+ | 138 | 21.77 | 36.9 | | |
| **Educational level (Partner)** | | | | 50.6 | 0.026 |
| <Middle/JSS/JHS | 189 | 31.3 | 61.1 | | |
| Middle/JSS/JHS | 123 | 43.8 | 14.9 | | |
| >Middle/JSS/JHS | 126 | 24.8 | 24 | | |
| **Occupation (Partner)** | | | | 58.4 | 0.0012 |
| Farmer/Agriculture | 187 | 28.3 | 61.7 | | |
| Government Worker | 56 | 11.4 | 14.2 | | |
| Trader/Artisans | 190 | 59.7 | 23.9 | | |
| Other | 5 | 0.6 | 0.2 | | |
| **Duration of marriage** | | | | 8.9 | 0.181 |
| <3 | 82 | 16.3 | 14.3 | | |
| 3–9 | 242 | 67.9 | 57.2 | | |
| 10+ | 114 | 15.8 | 28.5 | | |
| **Monthly Household Income** | | | | 69.47 | 0.001 |

*(Continued)*

**Table 6.** (Continued)

| Variables | Number N = 438 | Poor | Good | Ch12 | p-value |
|---|---|---|---|---|---|
| <100 | 156 | 16.5 | 57.8 | | |
| 100–999 | 243 | 78.1 | 38.4 | | |
| 1000+ | 39 | 5.4 | 37.7 | | |
| **Wealth Index** | | | | 6.3 | 0.57 |
| Lowest | 88 | 20.2 | 15.3 | | |
| second | 90 | 29.5 | 18.8 | | |
| middle | 85 | 18.7 | 23.9 | | |
| fourth | 88 | 32 | 26.5 | | |
| highest | 87 | 9.6 | 15.5 | | |
| **Residence** | | | | 0.026 | 0.949 |
| Urban | 193 | 79.6 | 80.3 | | |
| Rural | 245 | 20.4 | 19.7 | | |
| **Means of Transport** | | | | 19.3 | 0.02 |
| Boat | 1 | 0.2 | 0 | | |
| Motor bike | 164 | 38.7 | 28.6 | | |
| Vehicle | 63 | 13.8 | 4.9 | | |
| Walking | 210 | 47.3 | 66.4 | | |
| **Estimate travel time to nearest health facility** | | | | 0.23 | 0.851 |
| <30mins | 267 | 66.5 | 68.7 | | |
| >29mins | 171 | 33.6 | 31.3 | | |
| **Estimated distance to nearest health facility** | | | | 8.15 | 0.326 |
| <5km | 184 | 42.8 | 29.3 | | |
| 5-9km | 162 | 42.8 | 53.1 | | |
| 10+km | 92 | 14.4 | 17.7 | | |
| **Gestational Age at Booking for ANC** | | | | 32.8 | 0.0687 |
| 1st Trimester | 282 | 47.8 | 74.8 | | |
| 2nd Trimester | 131 | 45.6 | 23.2 | | |
| 3rd Trimester | 23 | 6.5 | 2 | | |
| **NHIS enrollment Status** | | | | 0.53 | 0.509 |
| Not enrolled | 30 | 9.1 | 7.1 | | |
| Enrolled | 408 | 10.9 | 92.9 | | |
| **Validity of NHIS card** | | | | 13.15 | 0.001 |
| Not active/Not enrolled | 217 | 58.4 | 40.3 | | |
| Active | 221 | 41.6 | 59.7 | | |
| **Gravidity** | | | | 0.45 | 0.928 |
| <3 | 208 | 44.1 | 44.7 | | |
| 3–4 | 137 | 32.6 | 29.8 | | |
| 5+ | 93 | 23.3 | 25.5 | | |
| **Parity** | | | | 4.3 | 0.589 |
| 0–2 | 237 | 52.5 | 55.8 | | |
| 3–4 | 124 | 21.9 | 26.2 | | |
| 5+ | 77 | 26 | 18.2 | | |
| **Number of Livebirths** | | | | 4.9 | 0.554 |
| 0–2 | 236 | 52.2 | 56.6 | | |
| 3–4 | 126 | 22 | 25.9 | | |
| 5+ | 76 | 25.7 | 17.5 | | |
| **Number of ANC Visits** | | | | 91.7 | 0.0012 |

*(Continued)*

**Table 6.** (Continued)

| Variables | Number N = 438 | Poor | Good | Ch12 | p-value |
|---|---|---|---|---|---|
| <4 | 100 | 49.8 | 9.3 | | |
| 4–7 | 264 | 43.5 | 81.4 | | |
| 8+ | 74 | 6.6 | 9.3 | | |
| **Gestational age at booking for IPTp-SP** | | | | 17.6 | 0.127 |
| <17 wks | 221 | 51.5 | 71.6 | | |
| 17-24wks | 145 | 30.9 | 18.5 | | |
| 25wks | 72 | 17.7 | 9.9 | | |

N: represents frequency, p<0.05

IPTp-SP during their ANC visits. SP is safe when given in the second and third trimester but this knowledge is lacking among the women [34]. Lack of knowledge of SP as medicine for IPTp underscores the need to create more awareness and improve specific knowledge on IPTp-SP among women of childbearing age. Therefore, changing provider practices at ANC clinics in the delivery of IPTp-SP services, supported by community awareness campaigns to educate mothers on the importance and benefits of IPT-SP usage, will most likely increase knowledge and improve uptake as demonstrated in some studies [35].

## Limitations

Geographically, the scope of the study was going to cover only the Volta Region (Volta and Oti) of Ghana, specifically the two purposively selected districts (North Tongu and Nkwanta North). In terms of the study unit, the scope was also narrowed to women who had given birth

**Table 7. Crude and adjusted associations between predictor variables and attitude service users.**

| Variables | Crude OR | 95%CI | p-value | Adjusted OR | 95%CI | p-value |
|---|---|---|---|---|---|---|
| **ANC visits** | | | | | | |
| <4 | 1 | | | 1 | | |
| 4–7 | 10.1 | 2.5–40.1 | 0.003 | 6.7 | 1.2–35.8 | 0.028 |
| 8+ | 7.6 | 2.5–22.4 | 0.002 | 17.2 | 3.4–87.3 | 0.002 |
| **District** | | | | | | |
| Nkwanta North | 1 | | | 1 | | |
| North Tongu | 0.1 | 0.1–0.2 | 0.000 | 0.1 | 0.02–0.18 | 0.000 |
| **Marital status** | | | | | | |
| single | 1 | | | 1 | | |
| married | 2.5 | 1.2–4.9 | 0.012 | 0.8 | 0.3–2.3 | 0.009 |
| others | 0.7 | 0.1–3.1 | 0.57 | 0.6 | 0.1–2.9 | 0.295 |
| **Religion** | | | | | | |
| Christians | 1 | | | 1 | | |
| Muslims | 1.01 | 0.2–5 | 0.98 | 0.8 | 0.3–2.4 | 0.643 |
| others | 1.01 | 0.4–2.4 | 0.982 | 0.3 | 0.1–0.7 | 0.015 |

**Crude odds ratio (OR)**: odds ratio of one independent variable predicting the dependent variable

**Adjusted odds ratio (OR)**: holds other relevant variables constant and provides the odds ratio from the potential variable of interest which is adjusted for the other independent variables included in the model. CI: confidence interval, p<0.05.

in the past 24 months. The focus of the study was on IPTp-SP policy and not on any other malaria intervention. Another possible limitation came from the use of questionnaires. The reliability of the data from the questionnaire depended on the skills and experience of field enumerators RAs that conducted the interviews as well as the extent to which the respondents could remember facts and events. Also, only well-trained graduates were used in the data collection to reduce errors. Furthermore, the study had financial challenges, hence the choice of only two districts for the study in the Volta Region. What this means is that the findings of the study should be interpreted with caution because the two participating districts were purposively selected based on low and high coverage of IPTp-SP [36].

## Conclusion

The findings from the present studies highlighted important factor such as number of antenatal visits that affect both knowledge of services and attitude to use IPTp-SP. Therefore, a community-based health promotion to increase knowledge on the benefit of IPTp-SP and improve attitude on timely and regular antenatal attendance should be encouraged. Further research needs to be carried out to understand the provider-side predictors influencing the knowledge and attitude of using 3+ doses of IPTp.

## Supporting information

**S1 Checklist. Questionnaire for service users (Survey).**
(DOCX)

## Acknowledgments

We would like to express our gratitude to the data collectors and the study participants for their time and dedication.

## Author Contributions

**Conceptualization:** Livingstone Asem.

**Data curation:** Livingstone Asem.

**Formal analysis:** Livingstone Asem.

**Funding acquisition:** Livingstone Asem.

**Methodology:** Livingstone Asem, Patrick Opoku Assuming.

**Resources:** Livingstone Asem.

**Software:** Livingstone Asem.

**Supervision:** Abdul-Gafaru Abdulia, Patrick Opoku Assuming, Gordon Abeka-Nkrumah.

**Validation:** Livingstone Asem.

**Visualization:** Livingstone Asem, Abdul-Gafaru Abdulia, Patrick Opoku Assuming.

**Writing – original draft:** Livingstone Asem.

**Writing – review & editing:** Abdul-Gafaru Abdulia, Patrick Opoku Assuming, Gordon Abeka-Nkrumah.

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
