## [Decision Letter · Decision Letter 0]

12 Aug 2024

Knowledge, and attitude of service user of intermittent preventive treatment of malaria in pregnancy using sulfadoxine pyrimethamine in the Volta Region of Ghana

PONE-D-24-28278

Dear Dr. Asem,

We’re pleased to inform you that your manuscript has been judged scientifically suitable for publication and will be formally accepted for publication once it meets all outstanding technical requirements.

Kind regards,

Edison Arwanire Mworozi, M.D

Academic Editor

PLOS ONE

2. We note that your Data Availability Statement is currently as follows: All relevant data are within the manuscript and its Supporting Information files

Reviewers' comments:

Reviewer's Responses to Questions

**Comments to the Author**

1. Is the manuscript technically sound, and do the data support the conclusions?

Reviewer #1: Yes

2. Has the statistical analysis been performed appropriately and rigorously? 

Reviewer #1: Yes

3. Have the authors made all data underlying the findings in their manuscript fully available?

Reviewer #1: Yes

4. Is the manuscript presented in an intelligible fashion and written in standard English?

Reviewer #1: Yes

5. Review Comments to the Author

**Reviewer #1:** Your study is a significant contribution to the field of maternal health and malaria prevention. It is well-written, logically structured, and provides a clear narrative that is easy to follow. The practical recommendations based on your findings have the potential to make a substantial impact on public health policy and interventions.

Thank you for your excellent work. I look forward to seeing the positive changes your research will inspire in the field.

Best regards,

6. PLOS authors have the option to publish the peer review history of their article (what does this mean?). If published, this will include your full peer review and any attached files.

Reviewer #1: **Yes: **Muhammad Ali

---

## [Editor Report · Acceptance letter]

27 Aug 2024

PONE-D-24-28278 

PLOS ONE

Dear Dr. Asem, 

I'm pleased to inform you that your manuscript has been deemed suitable for publication in PLOS ONE. Congratulations! Your manuscript is now being handed over to our production team.

Kind regards, 

on behalf of

Professor Edison Arwanire Mworozi 

Academic Editor

PLOS ONE